# On $1/n$ neural representation and robustness

**Josue Nassar**[*]
Department of Electrical and Computer Engineering
Stony Brook University
josue.nassar@stonybrook.edu

**Piotr Aleksander Sokol**[*]
Department of Neurobiology and Behavior
Stony Brook University
piotr.sokol@stonybrook.edu

**SueYeon Chung**
Center for Theoretical Neuroscience
Columbia University
sueyeon.chung@columbia.edu

**Kenneth D. Harris**
UCL Institute of Neurology
University College London
kenneth.harris@ucl.ac.uk

**Il Memming Park**
Department of Neurobiology and Behavior
Stony Brook University
memming.park@stonybrook.edu

## Abstract

Understanding the nature of representation in neural networks is a goal shared by neuroscience and machine learning. It is therefore exciting that both fields converge not only on shared questions but also on similar approaches. A pressing question in these areas is understanding how the structure of the representation used by neural networks affects both their generalization, and robustness to perturbations. In this work, we investigate the latter by juxtaposing experimental results regarding the covariance spectrum of neural representations in the mouse V1 (Stringer et al) with artificial neural networks. We use adversarial robustness to probe Stringer et al's theory regarding the causal role of a 1/n covariance spectrum. We empirically investigate the benefits such a neural code confers in neural networks, and illuminate its role in multi-layer architectures. Our results show that imposing the experimentally observed structure on artificial neural networks makes them more robust to adversarial attacks. Moreover, our findings complement the existing theory relating wide neural networks to kernel methods, by showing the role of intermediate representations.

## 1 Introduction

Artificial neural networks and theoretical neuroscience have a shared ancestry of models they use and develop, this includes the McCulloch-Pitts model [33], Boltzmann machines [1] and convolutional neural networks [16, 31]. The relation between the disciplines, however, goes beyond the use of cognate mathematical models and includes a diverse set of shared interests – importantly, the overlap in interests increased as more theoretical questions came to the fore in deep learning. As such, the two disciplines have settled on similar questions about the nature of 'representations' or neural codes: how they develop during learning; how they enable generalization to new data and new tasks; their dimensionality and embedding structure; what role attention plays in their modulation; how their properties guard against illusions and adversarial examples.

---

[*]equal contribution

Central to all these questions, is the exact nature of representations emergent in both artificial and biological neural networks. Even though relatively little is known about either, the known differences between both offer a point of comparison, that can potentially give us deeper insight into the properties of different neural codes, and their mechanistic role in giving rise to some of the observed properties. Perhaps the most prominent example of the difference between artificial and biological neural networks is the existence of adversarial examples [21, 23, 25, 44]– arguably they are of interest primarily not because of their genericity [14, 25], but because they expose the stark difference between computer vision algorithms and human perception.

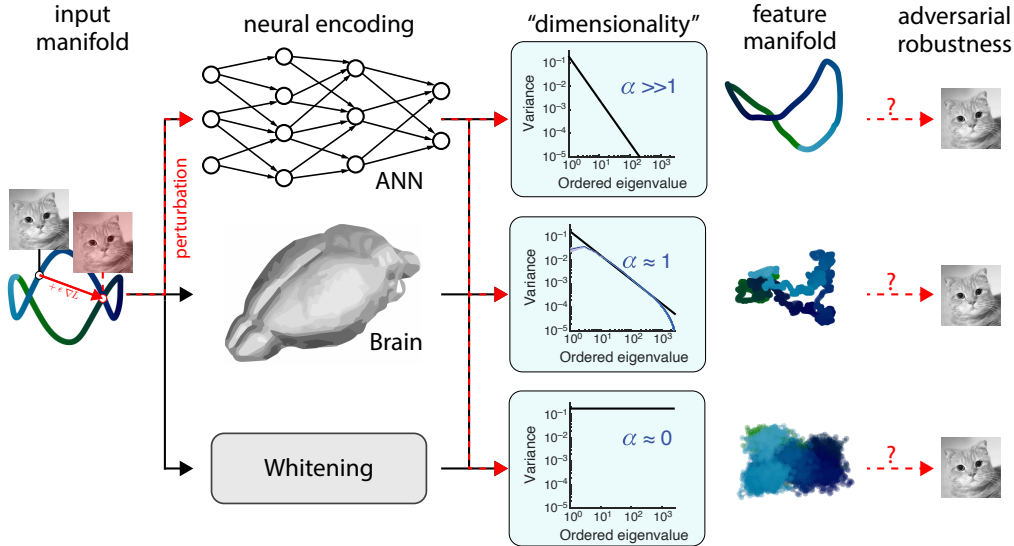

Figure 1: Studying the benefits of the spectrum of neural code for adversarial robustness. The neural code of the biological brain shows $1/n$ power-law spectrum and also robust. Meaningful manifolds in the input space can gain nonlinear features or lose their structure depending on the power-law exponent $\alpha$. Using artificial neural networks and statistical whitening, we investigate how the "dimensionality", controlled by $\alpha$, of the neural code impacts its robustness.

In this work, we use adversarial robustness to probe ideas regarding the 'dimensionality' of neural representations. The neuroscience community has advanced several normative theories, which include optimal coding [4], sparse coding [17, 35], as well a host of experimental data and statistical models [8, 18, 20, 22, 42] – resulting in, often conflicting, arguments in support of the prevalence of both low-dimensional and high-dimensional neural codes. By comparison, the machine learning community inspected the properties of hidden unit representations through the lens of statistical learning theory [15, 34, 47], information theory [40, 46], and mean-field and kernel methods [12, 27, 36]. The last two, by considering the limiting behavior as the number of neurons per layer goes to infinity, have been particularly successful, allowing analytical treatment of optimization, and generalization [2, 6, 7, 27].

Paralleling this development, a recent study recorded from a large number of mouse early visual area (V1) neurons [43]. The statistical analysis of the data leveraged kernel methods, much like the mean-field methods mentioned above, and has revealed that the covariance between neurons (marginalized over input) had a spectrum that decayed as a $1/n$ power-law regardless of the input image statistics. To provide a potential rationale for the representation to be poised between low and high dimensionality, the authors developed a corresponding theory that relates the spectrum of the neural repertoire to the continuity and mean-square differentiability of a manifold in the input space [43]. Even with this proposed theory, the mechanistic role of the $1/n$ neural code is not known, but Stringer et al. [43] conjectured that it strikes a balance between expressivity and robustness. Moreover, the proposed theory only investigated the relationship between the input and output of the neural network; as many neural networks involve multiple layers, it is not clear how the neural code used by the intermediate layers affects the network as a whole. Similarly existing literature that relates the spectra of kernels to their out-of-sample generalization implicitly treats multi-layer

architectures as shallow ones [5, 6, 10]. It is therefore desirable that a comprehensive theory ought to explain the role that the covariance spectrum plays at each layer of a neural network.

In this work, we empirically investigate the advantages of an $1/n$ neural code, by enforcing the spectral properties of the biological visual system in artificial neural networks. To this end, we propose a spectral regularizer to enforce a $1/n$ eigenspectrum.

With these spectrally-regularized models in hand, we aim to answer the following questions:

- Does having an $1/n$ neural code make the network more robust to adversarial attacks?
- For multi-layer networks, how does the neural code employed by the intermediate layers affect the robustness of the network?

The paper is organized as follows: we first provide a brief review of the empirical and theoretical results of Stringer et al. [43], followed by background information on deep neural networks. We then propose a spectral regularization technique and employ it in a number of empirical experiments to investigate the role of the $1/n$ neural code.

## 2   Background: $1/n$ Neural Representation in Mouse Visual Cortex

In this section, we briefly recap results from Stringer et al. [43]. To empirically investigate the neural representation utilized in the mouse visual cortex, the authors recorded neural activity from $\sim 10{,}000$ neurons while the animal was presented with large sets of images, including $2{,}800$ images from ImageNet [39]. They observed that the eigenspectrum of the estimated covariance matrix of the neural activity follow a power-law, i.e. $\lambda_n \propto n^{-\alpha}$ regardless of the input statistics, where for natural images a universal exponent of $\alpha \approx 1$ was observed.

The authors put forward a corresponding theory as a potential rationale for the existence of the $1/n$ spectra; importantly, the properties of the representation were investigated in the asymptotic regime of the number of neurons tending to infinity. Let $s \in \mathbb{R}^{d_s}$ be an input to a network, where $p(s)$ is supported on a manifold of dimension $d \leq d_s$, and let $x \in \mathbb{R}^N$ be the corresponding neural representation formed by a nonlinear encoding $f$, i.e. $x = f(s)$. Let $\lambda_1 \geq \lambda_2 \geq \cdots \geq \lambda_N$ be the ordered eigenvalues of $\mathrm{cov}(f(s))$. Under these assumptions Stringer et al. [43] proved that as $N \to \infty$, $\lambda_n$ must decay faster than $n^{-\alpha}$ where $\alpha = 1 + 2/d$ for $f$ to be continuous and differentiable, i.e. $f$ locally preserves the manifold structure.

While continuity and differentiability are desirable properties, the advantages of a representation with eigenspectrum decaying slightly faster than $n^{-1}$ is not apparent. Moreover, the theory abstracts away the intermediate layers of the network, focusing on the properties of $f$ but not its constituents in a multi-layered architecture where $f = f_1 \circ \cdots \circ f_D$. To investigate further, we turn to deep neural networks as a testbed.

## 3   Spectrally regularized Deep Neural Networks

Consider a feed-forward neural network with input, $s \in \mathbb{R}^{d_s}$, $D$ layers of weights, $\mathbf{W}^1, \ldots, \mathbf{W}^D$, biases, $b^1, \ldots, b^D$, and $D$ layers of neural activity, $x^1, \ldots, x^D$, where $x^l \in \mathbb{R}^{N_l}$, $\mathbf{W}^l \in \mathbb{R}^{N_l \times N_{l-1}}$, and $b^l \in \mathbb{R}^{N_l}$. The neural activity is recursively defined as,

$$x^l = f_l(x^{l-1}) \coloneqq \phi\left(\mathbf{W}^l x^{l-1} + b^l\right), \quad \text{for } l = 1, \cdots, D, \tag{1}$$

where $x^0 = s$, $\phi(\cdot)$ is an element-wise non-linearity and $b^l \in \mathbb{R}^{N_l}$ is a bias term. We define the mean and covariance of the neural activity at layer $l$ as $\mu^l = \mathbb{E}[x^l]$ and $\mathbf{\Sigma}^l = \mathbb{E}[(x^l - \mu^l)(x^l - \mu^l)^\top]$, respectively. Note that the expectation marginalizes over the input distribution, $p(s)$, which we assume has finite mean and variance, i.e. $\mu^0 < \infty$ and $\mathrm{Tr}(\mathbf{\Sigma}^0) < \infty$, where $\mathrm{Tr}(\cdot)$ is the trace operator.

To analyze the neural representation of layer $l$ we examine the eigenspectrum of its covariance matrix, $\mathbf{\Sigma}^l$, denoted by the ordered eigenvalues $\lambda_1^l \geq \lambda_2^l \geq \cdots \geq \lambda_{N_l}^l$, and corresponding eigenvectors $v_1^l, \ldots, v_{N_l}^l$. While the theory developed by Stringer et al. [43] dealt with infinitely wide networks, in reality, both biological and artificial neural networks are of finite width; although, empirical evidence

has shown that the consequences of infinitely wide networks are still felt by finite-width networks that are sufficiently wide [27, 36].

## 3.1 Spectral regularizer

In general, the distribution of eigenvalues of a deep neural network (DNN) is intractable and a priori there is no reason to believe it should follow a power-law — indeed, it will be determined by architectural choices, such as initial weight distribution and non-linearity, but also by the entire trajectory in parameter space traversed during optimization [27, 36]. A simple way to enforce a power-law decay without changing its architecture is to use the finite-dimensional embedding and directly regularize the eigenspectrum of the neural representation used at layer $l$. To this end we introduce the following regularizer:

$$R_l(\lambda_1^l, \ldots, \lambda_{N_l}^l) = \frac{\beta}{N_l} \sum_{n \geq \tau}^{N_l} \Big( (\lambda_n^l/\gamma_n^l - 1)^2 + \max(0, \lambda_n^l/\gamma_n^l - 1) \Big), \tag{2}$$

where $\gamma_n^l$ is a target sequence that follows a $n^{-\alpha_l}$ power-law, $\tau$ is a cut-off that dictates which eigenvalues should be regularized and $\beta$ is a hyperparameter that controls the strength of the regularizer. To construct $\gamma_n^l$, we create a sequence of the form $\gamma_n^l = \kappa n^{-\alpha_l}$ where $\kappa$ is chosen such that $\lambda_\tau^l = \gamma_\tau^l$. Since $\gamma_\tau^l = \lambda_\tau^l$, the ratio $\lambda_n^l/\gamma_n^l$ for $n \geq \tau$ serves as a proxy measure to compare the rates of decay between the eigenvalues, $\lambda_n^l$, and the target sequence, $\gamma_n^l$. Leveraging this ratio, $(\lambda_n^l/\gamma_n^l - 1)^2$ penalizes the network for using neural representations that stray away from $\gamma_n^l$; we note that this term equally penalizes a ratio that is greater than or less than 1. Noting that having a slowly decaying spectrum, $\lambda_n^l/\gamma_n^l > 1$, leads to highly undesirable properties (viz. discontinuity and unbounded gradients in the infinite-dimensional case), $\max(0, \lambda_n^l/\gamma_n^l - 1)$ is used to further penalize the network for having a spectrum that decays too slowly.

## 3.2 Training scheme

Naive use of (2) as a regularizer faces practical difficulty as it requires estimating the eigenvalues of the covariance matrix for each layer $l$ of each mini-batch, which has a computational complexity of $\mathcal{O}(N_l^3)$. Obtaining a reasonable estimate of $\lambda_n^j$ also requires a batch size at least as large as the widest layer in the network [11]. While the second issue is unavoidable, we propose a work around for the first one.

Performing an eigenvalue decomposition of $\boldsymbol{\Sigma}^l$ gives

$$\boldsymbol{\Sigma}^l = \mathbf{V}_l \boldsymbol{\Lambda}^l \mathbf{V}_l^\top, \tag{3}$$

where $\mathbf{V}_l$ is an orthonormal matrix of eigenvectors and $\boldsymbol{\Lambda}^l$ is a diagonal matrix with the eigenvalues of $\boldsymbol{\Sigma}^l$ on the diagonal. Using $\mathbf{V}_l$ we can diagonalize $\boldsymbol{\Sigma}^l$ to obtain

$$\mathbf{V}_l^\top \boldsymbol{\Sigma}^l \mathbf{V}_l = \boldsymbol{\Lambda}^l. \tag{4}$$

It's evident from (4) that given the eigenvectors, we could easily obtain the eigenvalues. Thus, we propose the following approach: at the beginning of each epoch an eigenvalue decomposition is performed on the full training set and the eigenvectors, $\mathbf{V}_l$ for $l = 1, \cdots, D$, are stored. Next, for each mini-batch we construct $\boldsymbol{\Sigma}^l$ and compute

$$\mathbf{V}_l^\top \boldsymbol{\Sigma}^l \mathbf{V}_l = \hat{\boldsymbol{\Lambda}}^l, \tag{5}$$

and the diagonal elements of $\hat{\boldsymbol{\Lambda}}^l$ are taken as an approximation for the true eigenvalues and used to evaluate (2). This approach is correct in the limit of vanishingly small learning rates [45].

When using the regularizer for training, the approximate eigenvectors are fixed and gradients are not back-propagated through them. Similar to batch normalization [26], the gradients are back-propagated through the construction of the empirical covariance matrix, $\Sigma^l$.

## 4 Experiments

To empirically investigate the benefits of a power-law neural representation and of the proposed regularization scheme, we train a variety of models on MNIST [30]. While MNIST is considered

a toy dataset for most computer vision tasks, it is still a good test-bed for the design of adversarial defenses [41]. Moreover, running experiments on MNIST has many advantages: 1) its simplicity makes it easy to design and train highly-expressive DNNs without relying on techniques like dropout or batch-norm; and 2) the models were able to be trained using a small learning rate, ensuring the efficacy of the training procedure detailed in section 3.2. This allows for the isolation of the effects of a $1/n$ neural representation, which may not have been possible if we used a dataset like CIFAR-10.

Recall that the application of Stringer et al. [43]'s theory requires an estimate of the manifold dimension, $d$, which is not known for MNIST. However, we make the simplifying assumption that it is sufficiently large such that $1 + {}^2/d \approx 1$ holds. For this reason we set $\alpha_l = 1$ for all experiments. Based on the empirical observation of Stringer et al. [43], we set $\tau = 10$ as they observed that the neural activity of the mouse visual cortex followed a power-law approximately after the tenth eigenvalue. For all experiments three different values of $\beta$ were tested, $\beta \in \{1, 2, 5\}$, and the results of the best one are shown in the main text (results for all values of $\beta$ are deferred to the appendix). The networks were optimized using Adam [29], where a learning rate of $10^{-4}$ was chosen to ensure the stability of the proposed training scheme. For each network, the batch size is chosen to be 1.5 times larger than the widest layer in the network. All results shown are based on 3 experiments with different random seeds[2].

The robustness of the models are evaluated against two popular forms of adversarial attacks:

- Fast gradient sign method (FGSM): Given an input image, $s_n$, and corresponding label, $y_n$, FGSM [23] produces an adversarial image, $\tilde{s}_n$, by

$$\tilde{s}_n = s_n + \epsilon \operatorname{sign}(\nabla_{s_n} L(f(s_n; \theta), y_n)), \tag{6}$$

  where $L(\cdot, \cdot)$ is the loss function and $\operatorname{sign}(\cdot)$ is the sign function.

- Projected gradient descent (PGD): Given an input image, $s_n$, and corresponding label, $y_n$, PGD [32] produces an adversarial image, $\tilde{s}_n$, by solving the following optimization problem

$$\arg\max_{\boldsymbol{\delta}} L(f(s_n + \boldsymbol{\delta}; \boldsymbol{\theta}), y_n),$$
$$\text{such that} \quad \|\boldsymbol{\delta}\|_\infty \leq \epsilon. \tag{7}$$

  To approximately solve (7), we use 40 steps of projected gradient descent

$$\boldsymbol{\delta} \leftarrow \operatorname{Proj}_{\|\boldsymbol{\delta}\|_\infty \leq \epsilon} \left( \boldsymbol{\delta} + \eta \nabla_{\boldsymbol{\delta}} L(f(s_n + \boldsymbol{\delta}; \boldsymbol{\theta}), y_n) \right), \tag{8}$$

  where we set $\eta = 0.01$ and $\operatorname{Proj}_{\|\boldsymbol{\delta}\|_\infty \leq \epsilon}(\cdot)$ is the projection operator.

The models are also evaluated on white noise corrupted images though we defer these results to the appendix.

To attempt to understand the features utilized by the neural representations, we visualize

$$J_n(x_j) = \frac{\partial \lambda_n^D}{\partial x_j}. \tag{9}$$

where $\lambda_n^D$ is the $n$th eigenvalue of $\Sigma^D$. To compute (9), the full data set is passed through the network to obtain $\Sigma^D$. The eigenvalues are then computed and gradients are back-propagated through the operation to obtain (9).

## 4.1 Shallow Neural Networks

To isolate the benefits of a $1/n$ neural representation, we begin by applying the proposed spectral regularizer on a sigmoidal neural network with one hidden layer of $N_1 = 2,000$ neurons with batch norm [26] (denoted by SpecReg) and examine its robustness to adversarial attacks. As a baseline, we compare it against a vanilla (unregularized) network.

Figure 2A, demonstrates the efficacy of the proposed training scheme as the spectra of the regularized network follows $1/n$ pretty closely. Figures 2B and C demonstrate that a spectrally regularized network is significantly more robust than it's vanilla counterpart against **both** FGSM and PGD attacks.

While the results are nowhere near SOTA, we emphasize that this robustness was gained *without training on a single adversarial image, unlike other approaches*. The spectrally regularized network has a much higher effective dimension (Fig. 2A), and it learns a more relevant set of features as indicated by the sensitivity maps (Fig. 2D). We note that while the use of batch norm helped to increase the effectiveness of the spectral regularizer, it did not affect on the robustness of the vanilla network. In the interest of space, the results for the networks without batch norm are in the appendix.

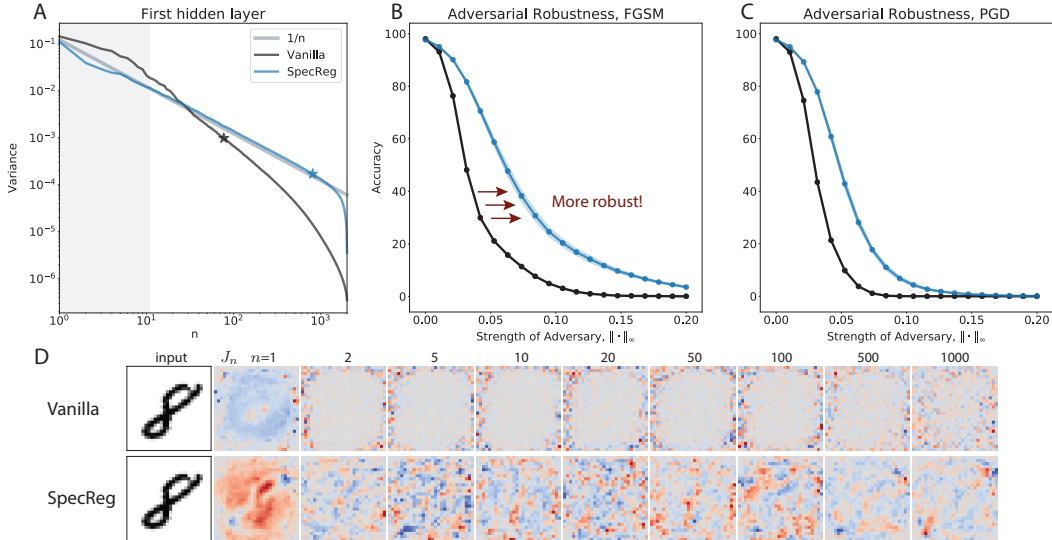

Figure 2: A $1/n$ neural representation leads to more robust networks. A) Eigenspectrum of a regularly trained network, Vanilla, and a spectrally regularized network, SpecReg. The shaded grey area are the eigenvalues that are not regularized. Star indicates the dimension at which the cumulative variance exceeds 90%. B & C) Comparison of adversarial robustness between vanilla and spectrally regularized networks where the shaded region is $\pm$ 1 standard deviation computed over 3 random seeds. D) The sensitivity of $\lambda_n$ with respect to the input image.

## 4.2 Deep Neural Networks

Inspired by the results in the previous section, we turn our attention to deep neural networks. Specifically, we experiment on a multi-layer perceptron (MLP) with three hidden layers where $N_1 = N_2 = N_3 = 1,000$ and on a convolutional neural network (CNN) with three hidden layers: a convolutional layer with 16 output channels with a kernel size of (3, 3), followed by another convolutional layer with 32 output channels with a kernel of size (3, 3) and a fully-connected layer of width 1,000 neurons, where max-pooling is applied after each convolutional layer. To regularize the neural representation utilized by the convolutional layers, the output of all the channels is flattened together and the spectrum of their covariance matrix is regularized. We note that no other form of regularization is used to train these networks i.e. batch norm, dropout, weight decay, etc.

We demonstrate empirically the importance of intermediate layers in a deep neural network, as networks with "bad" intermediate representations are shown to be extremely brittle.

### 4.2.1 The Importance of Intermediate Layers in Deep Neural Networks

Theoretical insights from Stringer et al. [43] as well as other works [6, 27] do not prescribe how the spectrum of intermediate layers should behave for a "good" neural representation. Moreover, it is not clear how the neural representation employed by the intermediate layers will affect the robustness of the overall network. The theory of Stringer et al. [43] suggests that the intermediate layers of the network do not matter as long as the neural representation of the last hidden layer is $1/n$.

To investigate the importance of the intermediate layers, we "break" the neural representation of the second hidden layer by whitening its neural activity

$$\tilde{x}^2 = R_2^{-1}(x^2 - \mu^2) \tag{10}$$

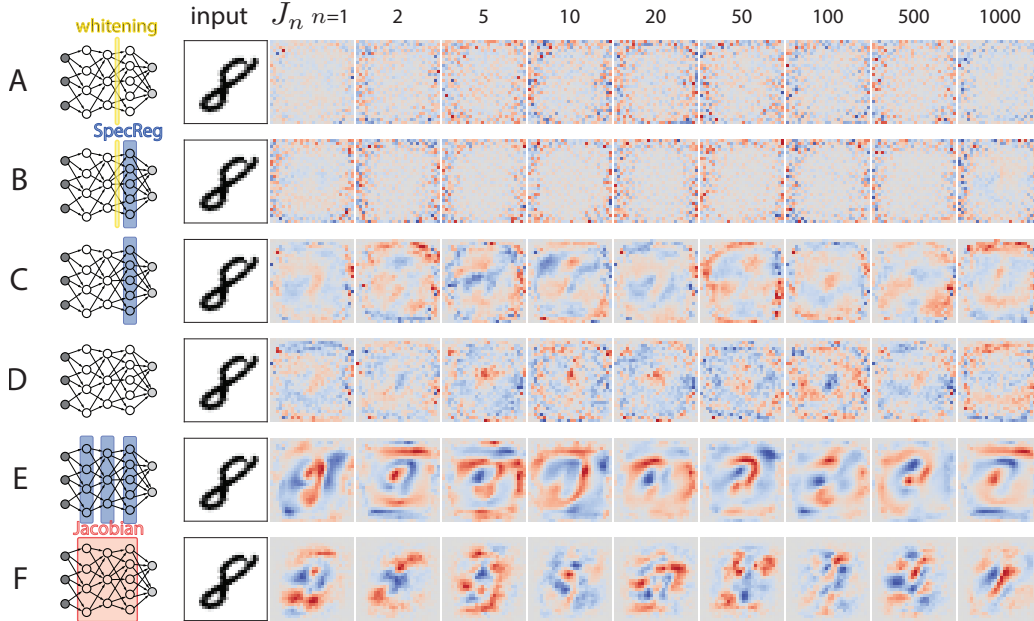

Figure 3: Sensitivity maps for 3-layer multi-layer perceptrons. Each row corresponds to a different experiment: (A) MLP with whitening layer, (B) SpecReg only on the last layer after whitening, (C) SpecReg only on the last layer, (D) Vanilla MLP, (E) SpecReg on every layer, (F) Jacobian regularization. Each sensitivity image corresponds to the $n$-th eigenvalue on the last hidden layer.

where $\boldsymbol{R}_2$ is the Cholesky decomposition of $\boldsymbol{\Sigma}^2$, leading to a flat spectrum which is the worst case scenario under the asymptotic theory in Stringer et al. [43]. To compute (10), the sample mean, $\hat{\boldsymbol{\mu}}^2$, and covariance, $\hat{\boldsymbol{\Sigma}}^2$, are computed for each mini-batch. Training with the whitening operation is handled similarly to batch norm [26], where gradients are back-propagated through the sample mean, covariance and Cholesky decomposition.

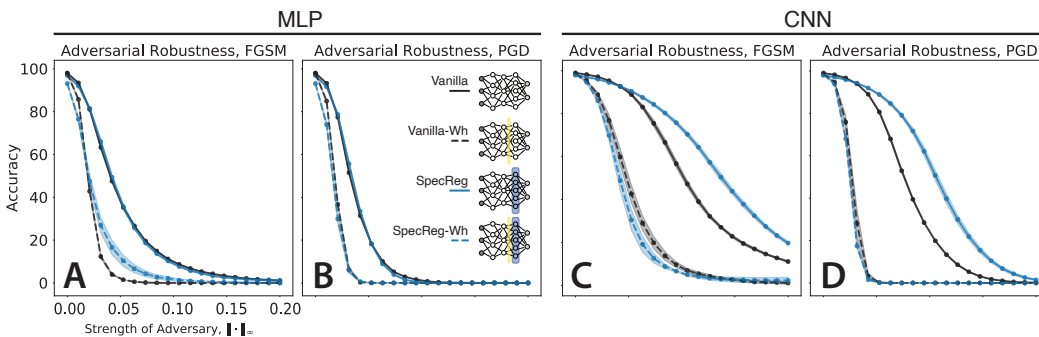

Figure 4: Spectral regularization does not rescue robustness lost by whitening the intermediate representation. (A,B) For MLP, adversarial robustness for FGSM and PGD indicates a more fragile code resulting from whitening. Spectral regularization of the last hidden layer does not improve robustness. (C,D) For CNNs, spectral regularization of the last layer enhances robustness, but for MLPs it does not. Same conventions as Fig. 2B,C and Fig. 3.

When the intermediate layer is whitened, the resulting sensitivity features lose structure (Fig. 3A compared to Fig. 3D). This network is less robust to adversarial attacks (Fig. 4A,B dashed black) consistent with the structureless sensitivity map.

We added spectral regularization to the last hidden layer in hopes of salvaging the network (see Sec. 3.2 for details). Although the resulting spectrum of the last layer shows a $1/n$ tail (Fig. A10), the robustness is not improved (Fig. 4A,B). Applying the asymptotic theory to the whitened output, the

neural representation is "bad" and the manifold becomes non-differentiable. Spectral regularization of the last hidden layer cannot further fix this broken representation even in a finite-sized network (Fig. 3B).

Interestingly, for the MLP, regularizing only the last hidden layer (without whitening) improved the sensitivity map (Fig. 3C) but had no effect on the robustness of the network (Fig. 4A,B), suggesting that a $1/n$ neural representation at the last hidden layer is not sufficient. In contrast, regularizing the last hidden layer of the CNN does increase the robustness of the network (Fig. 4C,D).

## 4.3 Regularizing all layers of the network increases adversarial robustness

The previous section showcased the importance of the intermediate representations in the network. The MLP results showcased that regularizing just the last hidden layer is not enough. Thus, we investigate the effect of spectrally regularizing all the hidden layers in the network. As an additional comparison, we train networks whose Jacobian is regularized [24, 28, 38]

$$\frac{\beta_j}{B} \sum_{n=1}^{B} \left\| \frac{\partial L(f(\boldsymbol{s}_n; \theta), y_n)}{\partial \boldsymbol{s}_n} \right\|_F^2 \tag{11}$$

where $(\boldsymbol{s}_n, y_n)$ is the $n^{\text{th}}$ training data point, $B$ is the batch size and $\beta_j$ is the strength of the regularizer where we follow [24] and set $\beta_j = 0.01$. We compare against this type of regularization as it is one of

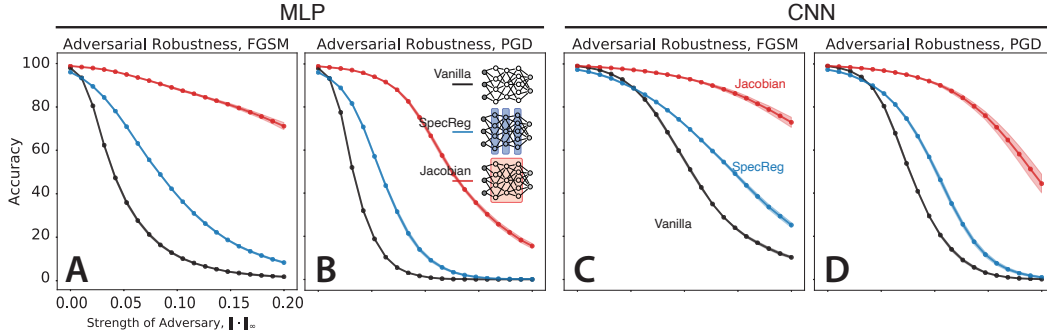

Figure 5: Spectral regularization of all layers improves robustness in 3-layer MLP and CNN. Same conventions as Fig. 4.

the few approaches that do not rely on training on adversarial examples nor does it require changing inputs, nor models. The spectra of the networks are in the appendix.

Regularizing all the layers of the MLP leads to an increase in robustness, as is evident in figures 5A,B. The neural representation induced by the regularizer favors *smoother*, more global features. This characteristic is shared both by our proposed regularization technique as well as Jacobian regularization, as can be seen in figure 3. Interestingly, these observations find a counterpart in existing results on adversarial robustness in CNNs [48].

While the regularizer was able to enforce a $1/n$ decay for the last hidden layer of the network, its effect was less pronounced at earlier layers where it was observed that that the networks preferred to relax towards $1/n$ as opposed to every hidden layer having a $1/n$ neural representation. This suggests an alternative approach where we slowly decrease $\alpha$ over depth, $\alpha_1 \geq \alpha_2 \geq \cdots \alpha_D = 1$; we leave this for future work. While regularizing all the layers in the CNN leads to an increase in robustness figure 5C,D, the effect is comparable to just regularizing the last layer (Fig. 4C,D). We note that regularizing the spectra is inferior to regularizing the Jacobian in terms of adversarial robustness. Interestingly, when evaluating the robustness of the networks on white noise corrupted images, the gap in performance between the Jacobian regularized and the spectrally regularized networks decreased (Fig. A21, A25) where for the CNNs, the spectrally regularized networks performed better than the Jacobian regularized networks (Fig. A25). Examining the spectra in figures A18, A22, we see that the neural representation utilized by the Jacobian regularized networks do not follow $1/n$. Thus, while networks whose neural representation have a $1/n$ eigenspectrum are robust, robust networks do not necessarily possess a $1/n$ neural representation.

# 5 Discussion

In this study, we trained artificial neural networks using a novel spectral regularizer to further understand the benefits and intricacies of a $1/n$ spectra in neural representations. We note that our current implementation of the spectral regularization is not intended to be used in general but rather a straightforward embodiment of the study objective. As the result suggests, a general encouragement of $1/n$-like spectrum could be beneficial in wide neural networks, and special architectures could be designed to more easily achieve this goal. The results have also helped to elucidate the importance of intermediate layers in DNNs and may offer a potential explanation for why batch normalization reduces the robustness of DNNs [19]. Furthermore, the results contribute to a growing body of literature that analyzes the generalization of artificial neural networks from the perspective of kernel machines (viz [6, 27]). As mentioned before, the focus in those works is on the input-output mapping, which does away with the intricate structure of representations at intermediate layers. In this work, we take an empirical approach and probe how the neural code for different hidden layers contributes to overall robustness.

From a neuroscientific perspective, it is interesting to conjecture whether a similar power-law code with a similar exponent is a hallmark of canonical cortical computation, or whether it reflects the unique specialization of lower visual areas. Existing results in theoretical neuroscience point to the fact neural code dimensionality in visual processing is likely to be either transiently increasing and then decreasing as stimuli are propagated to downstream neurons, or monotonically decreasing [3, 9, 13]. Resolving the question of the ubiquity of power-law-like codes with particular exponents can therefore be simultaneously addressed in-vivo and in-silico, with synthetic experiments probing different exponents at higher layers in an artificial neural network. Moreover, this curiously relates to the observed, but commonplace spectrum flattening for random deep neural networks [36], and questions about its effect on information propagation. We leave these questions for future study.

## Broader Impact

Adversarial attacks pose threats to the safe deployment of AI systems– both safety from malicious attacks but also robustness that would be expected in intelligent devices such as self-driving cars [37] but also facial recognition systems. Our neuroscience-inspired approach, unlike widely use adversarial attack defenses, does not require generation of adversarial input samples, therefore it potentially avoids the pitfalls of unrepresentative datasets. Furthermore, our study shows possibilities for the improvement of artificial systems with insights gained from biological systems which are naturally robust. Conversely it also provides a deeper, mechanistic understanding of experimental data from the field of neuroscience, thereby advancing both fields at the same time.

## Acknowledgments and Disclosure of Funding

This work is supported by the generous support of Stony Brook Foundation's Discovery Award, NSF CAREER IIS-1845836, NSF IIS-1734910, NIH/NIBIB EB026946, and NIH/NINDS UF1NS115779. JN was supported by the STRIDE fellowship at Stony Brook University. SYC was supported by the NSF NeuroNex Award DBI-1707398 and the Gatsby Charitable Foundation. KDH was supported by Wellcome Trust grants 108726 and 205093. JN thanks Kendall Lowrey, Benjamin Evans and Yousef El-Laham for insightful discussions and feedback. JN also thanks the Methods in Computational Neuroscience Course at the Marine Biological Laboratory in Woods Hole, MA for bringing us all together and for the friends, teaching assistants, and faculty who provided insightful discussions, support and feedback.

## Footnotes

[2]Code is available at https://github.com/josuenassar/power_law

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
