[Supplementary Material]

# A    Additional Figures for Section 4.1

Figure A6: A) Eigenspectrum of a regularly trained network, Vanilla, and a spectrally regularized network, SpecReg, for various values of $\beta$. The shaded grey area are the eigenvalues that are not regularized. Star indicates the dimension at which the cumulative variance exceeds 90%. B & C) Comparison of adversarial robustness between vanilla and spectrally regularized networks where the shaded region is $\pm 1$ standard deviation computed over 3 random seeds.

Figure A7: Robustness against white noise corrupted images for single layer MLPs for various values of $\beta$ where the shaded region is $\pm 1$ standard deviation computed over 3 random seeds.

## A.1 Networks without batch-norm

For the single-layer MLPs, we found that the inclusion of batch-norm allows for the eigenspectrum of the networks to reach $1/n$ faster than their non-batch-norm counterparts.

Figure A8: A) Eigenspectrum of a regularly trained network, Vanilla, and a spectrally regularized network, SpecReg, for various values of $\beta$ where batch-norm is not used. The shaded grey area are the eigenvalues that are not regularized. Star indicates the dimension at which the cumulative variance exceeds 90%. B & C) Comparison of adversarial robustness between vanilla and spectrally regularized networks where the shaded region is $\pm$ 1 standard deviation computed over 3 random seeds.

Figure A9: Robustness against white noise corrupted images for single layer MLPs for various values of $\beta$ where the shaded region is $\pm$ 1 standard deviation computed over 3 random seeds.

# B    Additional Figures for Section 4.2

## B.1    MLP

Figure A10: Eigenspectrum of whitened and non-whitened MLPs. Diamond/star indicates the dimension at which the cumulative variance exceeds 90 % for whitened/non-whitened networks. A) Eigenspectrum of the first hidden layer. B) Eigenspectrum of the second hidden layer. Whitening the neural representation leads to an approximately flat spectra. C) Eigenspectrum of the third hidden layer. The shaded gray area are the eigenvalues that are not regularized.

Figure A11: Eigenspectrum of whitened and non-whitened MLPs for various values of $\beta$. Diamond/star indicates the dimension at which the cumulative variance exceeds 90 % for whitened/non-whitened networks A) Eigenspectrum of the first hidden layer. B) Eigenspectrum of the second hidden layer. Whitening the neural representation leads to an approximately flat spectra. C) Eigenspectrum of the third hidden layer. The shaded gray area are the eigenvalues that are not regularized.

Figure A12: Adversarial robustness of the MLP for various values of $\beta$ where the shaded region is $\pm$ 1 standard deviation computed over 3 random seeds.

Figure A13: Robustness against white noise corrupted images for the CNN for various values of $\beta$ where the shaded region is $\pm$ 1 standard deviation computed over 3 random seeds.

## B.2 CNN

Figure A14: Eigenspectrum of whitened and non-whitened CNNs. Diamond/star indicates the dimension at which the cumulative variance exceeds 90 % for whitened/non-whitened networks. A) Eigenspectrum of the first hidden layer. B) Eigenspectrum of the second hidden layer. Whitening the neural representation leads to an approximately flat spectra. C) Eigenspectrum of the third hidden layer. The shaded gray area are the eigenvalues that are not regularized.

Figure A15: Eigenspectrum of whitened and non-whitened CNNs for various values of $\beta$. Diamond/star indicates the dimension at which the cumulative variance exceeds 90 % for whitened/non-whitened networks A) Eigenspectrum of the first hidden layer. B) Eigenspectrum of the second hidden layer. Whitening the neural representation leads to an approximately flat spectra. C) Eigenspectrum of the third hidden layer. The shaded gray area are the eigenvalues that are not regularized.

Figure A16: Adversarial robustness of the CNN for various values of $\beta$ where the shaded region is $\pm$ 1 standard deviation over 3 random seeds.

Figure A17: Robustness against white noise corrupted images for the CNN for various values of $\beta$ where the shaded region is $\pm 1$ standard deviation computed over 3 random seeds.

## C  Additional Figures for Section 4.3

### C.1  MLP

Figure A18: Eigenspectrum of MLPs. Star indicates the dimension at which the cumulative variance exceeds 90%. The shaded grey area are the eigenvalues that are not regularized.

Figure A19: Eigenspectrum of MLPs for various values of $\beta$. Star indicates the dimension at which the cumulative variance exceeds 90%. The shaded grey area are the eigenvalues that are not regularized.

Figure A20: Adversarial robustness of the MLP for various values of $\beta$ where the shaded region is $\pm$ 1 standard deviation computed over 3 random seeds.

Figure A21: Robustness against white noise corrupted images for the MLP for various values of $\beta$ where the shaded region is $\pm$ 1 standard deviation computed over 3 random seeds.

## C.2 CNN

Figure A22: Eigenspectrum of CNNs. Star indicates the dimension at which the cumulative variance exceeds 90%. The shaded grey area are the eigenvalues that are not regularized.

Figure A23: Eigenspectrum of CNNs for various values of $\beta$. Star indicates the dimension at which the cumulative variance exceeds 90%. The shaded grey area are the eigenvalues that are not regularized.

Figure A24: Adversarial robustness of the CNN for various values of $\beta$ where the shaded region is $\pm$ 1 standard deviation computed over 3 random seeds.

Figure A25: Robustness against white noise corrupted images for the CNN for various values of $\beta$ where the shaded region is $\pm$ 1 standard deviation computed over 3 random seeds.