[Reviews · NeurIPS 2020]

Review 1

Summary and Contributions: This work begins by introducing a key new result from the neuroscience community: Stringer et al. 2019 [43] found that the eigenvalue spectrum of the estimated neural activity covariance matrix in the biological mouse V1 visual cortex decays like 1/n when presented with images from ImageNet. Stringer et al. further proved that this must be the case in order to preserve certain fundamental properties of the neural coding, however the practical advantages of a such a representation remain unclear. This work investigates a conjecture of Stringer that the 1/n neural code helps balance the expressivity and robustness of representations. To do so, a spectral regularizer is purposefully devised to enforce such a representation, and its efficacy is validated by experiments. It is found that the decaying eigenvalue spectrum improves robustness to FGSM and PGD perturbations on MNIST. It is further found that it is not enough to have a 1/n code at the final hidden layer for robustness, thus clarifying the importance of intermediate layers, which was absent in the analysis of Stringer.

Strengths: This work clarifies the relationship between the spectrum of the covariance of the hidden layers’ activations and adversarial robustness / interpretability of input-sensitivity maps. In addition to its relevance to the ML community, the work may be relevant to neuroscientists who wish to further probe the effects of a 1/n code in artificial networks. The work shows a considerable improvement in robustness to adversarial examples (for a range of perturbation sizes) without training on them. This is an exciting result as adversarial training is computationally expensive and does not usually generalize well to novel attacks. The proposed approach seems more likely to confer broader robustness to other threat models and corruptions. The paper is well focused and precise in scope.

Weaknesses: Note: this section is structured to seed further discussions, its length relative to the above "Strengths" does not imply that the Weaknesses outweigh the Strengths. As the authors mention, their novel spectral regularizer is not the main contribution of the work, and there likely isn’t enough comparison to the existing methods for spectral regularization for this to be taken on its own as a state-of-the-art regularizer for deep networks. The method also relies on a small learning rate as the eigenvectors of the activations’ covariance matrix are estimated once per epoch using the full training set and this snapshot is propagated to subsequent mini-batch calculations throughout the epoch. It would be nice to quantify/estimate the error introduced in this estimation due to the finite learning rate, otherwise it could be prohibitive to scale the method to more challenging sets like CIFAR that generally require training with large initial learning rates for competitive performance. The experiments that are based on MNIST with FGSM/PGD would not generally be considered as very exciting or convincing to the deep learning community, but the authors are right to point out that robustness remains essentially “unsolved” on MNIST (or more specifically that there are gaps in our understanding about the extent to which robustness can be achieved on MNIST). There is also a mismatch between MNIST <-> ImageNet used by Stringer. I think this can mostly be excused due to the novelty of the work which bridges two communities, but I would like to see some additional evaluation even in the Supplementary. It could be a strength that the theory generalizes to MNIST given the extent to which it differs from ImageNet and other natural scenes. Nonetheless, it would be a good idea to validate some of the main findings on a natural image dataset like CIFAR-10. Regarding the evaluation, I have no reason to suspect gradient issues to have interfered with the white-box attacks, which is a common concern anytime FGSM/PGD are used on their own. However, the paper would likely reach a broader audience if other natural notions of robustness, for example to the Common Corruptions suite (available now for MNIST, CIFAR-10, and ImageNet) were considered and would help prevent others’ concerns about gradient issues. On this point, the curve for Jacobian regularization seems quite high and never reaches zero accuracy. It would be good to confirm that this indeed reaches zero, possibly by breaking the x-axis to display, should this require a much larger perturbation. Additive white Gaussian noise is also a simple test for correctness.

Correctness: The proposed regularizer and implementation method seem reasonable and correct, aside from parts that are approximations and acknowledged as such. The evaluation is performed by generating curves for FGSM/PGD perturbations until zero accuracy is reached. This combined with the sensitivity visualizations convince me that gradient masking is not at play. It would be good to clarify if pixels were normalized to [0-1] for interpreting epsilon values. If I understood correctly, a generic whitening is used for the hidden layers of the deep networks, while batch normalization (BN) is only used for the shallow network. Why is this the case, i.e., why not be consistent and use one, or the other, or both?

Clarity: The work is very well written. A few points of clarification: Can you elaborate on the discussion point re how the results may explain why BN reduces robustness of DNNs? Is there a relationship between BN and the generic whitening procedure used? I would expect BN to yield similar sensitivity images as those shown for the whitening and vanilla case in Fig 2 given that it normalizes the activations inversely proportional to the original variance of each dimension or pixel. The border background area for MNIST with low original variance would be amplified relative to the foreground pixels with BN. I believe a reason BN did not affect the robustness of the shallow network could be due to the small learning rate (used to accommodate the proposed regularizer). A stronger vanilla baseline would be to use a large initial learning rate (1e-1, or 1e-2) with SGD, and optionally L1 regularization for max-norm Lp attacks, but it may be difficult to achieve competitive clean accuracy with this recipe. I have also found that the type of normalization has a big impact on MNIST robustness, i.e. per-image standardization versus per-pixel like BN. - I don’t believe the optimizer (e.g., SGD vs Adam) and mini-batch size used in the experiments were mentioned. Misc: - Line 151: Can you clarify the meaning of “best” wrt the selection of \beta? - Line 155: I would refer to FGSM and PGD as attack/perturbation algorithms versus threat models (i.e., white-box, black-box). Max-norm perturbations could be considered as an idealized threat model. - Line 165: typo “eigenvlaues” - Line 171: "unregularized network" could refer to unregularized with SpecReg or BN. Please clarify. - Line 234: Can you clarify “relax towards”? - Figure 2 – FGSM and PGD curves could be plotted on same axes, readers will want to compare these to ensure PGD achieves greater reduction in accuracy at each point since it’s similar to an iterative version of FGSM.

Relation to Prior Work: There is no section titled “Related Work”, but I found that references were well-cited throughout the paper. I can appreciate that there are a lot of works on spectral normalization or constraining Lipschitz constant that may not be strictly relevant to enforce the 1/n spectrum, but I think some of these should be discussed in a dedicated related work section. I believe some of these are complementary to the proposed method. For example: - Miyato et al., ICLR 2018 https://openreview.net/forum?id=B1QRgziT- - Tsuzuku et al., NeurIPS 2018 http://papers.nips.cc/paper/7889-lipschitz-margin-training-scalable-certification-of-perturbation-invariance-for-deep-neural-networks.pdf - Anil et al., ICML 2019 http://proceedings.mlr.press/v97/anil19a.html

Reproducibility: Yes

Additional Feedback: Note to authors, the submission file size is quite large at ~13MB. I believe this is due to the number of dots used to depict manifolds in Figure 1. There was enough of a delay rendering Fig. 1 on my machine that it seemed like an animation. I suspect that the number of dots could be reduced significantly without degrading the figure aesthetics to make the PDF more accessible. -------------- Update following the rebuttal -------------------- I appreciate the authors' rebuttal and am cognizant of the burden to run new experiments, especially during the global pandemic situation. Thanks for the clarification about whitening versus BN. One question that arose is that the mini-batch size used to train models and estimate the instantaneous eigenvalues for the regularizer was not mentioned (or at least we couldn't find it). Thankfully the authors provided detailed source code that showed a batch size of 1500-3500 for MLPs, and 6000 for CNNs. This should be mentioned in the main text, especially as these values are unusually large. I suspect this could also be a contributing factor making it difficult to scale the experiments to CIFAR-10 due to the batch size needing to be at least as large as the largest layer, and such large batch sizes potentially exceeding GPU memory. This might be something to discuss in the main text. I don't think we should ignore a method simply because it doesn't immediately apply to more complex datasets, but some additional discussion about scaling limitations will be useful for others that may attempt to extend the work, as well as to help the reader appreciate why the analysis was limited to a low resolution dataset.


Review 2

Summary and Contributions: This paper proposes a novel regularization scheme for artificial neural networks based on eigenspectrum-decay normalization for intermediate layers' activations. The approach is inspired by experimental results on the covariance spectrum of neural representations in a mouse. The proposed regularization scheme is studied in the context of robustness to adversarial attacks on the MNIST dataset. The approach achieves higher performance with respect to the "no-regularization" baseline, but is outperformed by Jacobian regularization techniques on the same task.

Strengths: I believe that one of the main strength of the approach consists in its underlying idea and motivation. In the opinion of the reviewer, the goal of understanding the intermediate representation in neural networks is a very important one and of great relevance for the neuroscience and machine learning sub-communities of NeurIPS. The idea of regularizing intermediate activations to improve adversarial robustness is clearly not novel, but the proposed methodology is new to me. In addition, it is extremely valuable to see that neuroscience experiments have implication in the design/optimization of artificial neural networks, in spite of the fact that the two fields have largely diverged in the past decade. Assuming that the visual cortex of a mouse follows the same behavior of intermediate representation of neural networks is probably oversimplifying, but could motivate future research in this direction. The claim are sound but are sometimes a bit too strong. First, the claim that the eigenspectrum generalization could be beneficial for general vision tasks should be a bit softened, since the experiments only show application for robustness to adversarial attacks (which is a missed opportunity by the way, why not analyzing the performance of the proposed methods for other tasks like segmentation, detection, optical flow prediction, etc.?). For the task of robustness against adversarial attacks the paper does not achieve comparable performance with existing methods. This observation should come way earlier in the paper (ideally in the introduction). Of course SOTA performance is not necessary for all publications, but it is important to clearly position this work as an instance of application of neuroscience findings to neural networks only, with (not yet) strong practical advantages/applications.

Weaknesses: I believe the main weaknesses of the paper to be in its experimental setup. First of all, the proposed method is only evaluated in the context of adversarial robustness, but could theoretically not be limited to this task. Why not applying the same exact methodology on different tasks, e.g. object segmentation, object detection, or optical flow prediction? How would the regularization scheme affect performance? Could it serve as a general normalization technique and not only provide some degree of adversarial robustness? Connected to this topic, would be very nice to have a formal discussion of the relation between the proposed approach and classical regularization techniques (like batchNorm, InstanceNorm, LayerNorm, etc.) as well as with other methods for adversarial robustness (e.g. Jacobian regularization). What is the effect on the eigen-spectrum when doing BN? What the effect when doing Jacobian smoothing? Answering these questions could offer lots of insights for improving the approach. As stated above, the approach does not achieve comparable performance to existing methods. This is not a problem, but it would have been better to offer some more intuitions to explain this behavior. For example, how is the performance affected by some important hyper-parameters, e.g. the depth of the network, the frequency and strength of regularization (as controlled by \beta and the number of regularized layers)? Also, it appears that Jacobian methods impose a decay of the eigen-spectrum which is significantly faster than 1/n. What would be the effect of increasing the decay from 1/n to 1/n^{\alpha} with \alpha > 1? On a minor note, I am a bit puzzled by the choice of regularization function (Eq. 2). Specifically, the paper focuses on the importance on the 1/n decay but then suddenly the loss function favors something faster than 1/n (due to the second loss term). Was the second term of Eq. 2 only added because of an empirical advantage? If yes, doesn't this suggest that faster decays could be beneficial?

Correctness: I think that the claims are overall clear and well defined. It should be clarified that the approach does not aim to get close to SOTA for robustness to adversarial attacks, and only tries to implement a neuroscience theory into ANNs by making the (over-simplifying?) assumption that ANNs behave like neurons in the visual cortex of a mouse.

Clarity: The paper is overall well written and easy to follow. I could only find a typo in the caption of Fig. 1. This figure could also be improved a bit. I was able to understand it only after reading the methodology/experimental setup (looks like an input is passed through an ANN, a Brain, and a "Whitening")...

Relation to Prior Work: The paper makes overall a good job in position itself with respect to previous work. However, as stated above, it would have been nicer to have a more detailed comparison/discussion with respect to other regularization functions and smoothing methods for adversarial robustness.

Reproducibility: Yes

Additional Feedback: I think that with some more tasks the paper would get significantly stronger, I would definitely encourage the authors to try them out. ------------ UPDATE ----------------- Thanks to the authors for the feedback on my questions. In light of the rebuttal and the discussion with other reviewers, my main concerns were addressed. As a result, I see this work as a good contribution for the community. I would however invite the authors to further clarify their claims in the introduction and the challenges to scale this idea to more complex datasets.


Review 3

Summary and Contributions: Thanks to the authors for their rebuttal. [ORIGINAL REVIEW] This paper investigates whether the 1/n power spectrum recently observed in mouse visual cortex is also observed in artificial neural networks, and whether inducing this particular power spectrum confers adversarial robustness.

Strengths: The paper clearly defines and motivates an interesting problem about the nature of neural representations. In particular, it draws connections to recent work by Stringer et al in the mouse visual cortex that observed that neural representations have a 1/n power spectrum. Stringer et al hypothesized that this could confer advantages in that the representation would be robust to small perturbations. This paper directly tests that hypothesis. I appreciated that the paper was clear, direct, and did not try to mask or massage results. In particular, I see this as a bit of a negative result in that it seems that a 1/n representation is neither necessary nor sufficient for adversarial robustness. However, I think the paper presents this result using clear methodology and motivation, which I appreciated. Side note: I would put something about this conclusion directly in the appendix, e.g. "For deeper networks, we found that a 1/n spectrum was neither necessary nor sufficient to confer adversarial robustness." The natural question one has after reading this is ... well what is a 1/n power spectrum good for then? Do the authors have any hypotheses about what advantages to look for next?

Weaknesses: The main weakness, for me, is that the only experiments are using the MNIST dataset. Having experiments with another dataset would not only increase confidence that these results are general, but I think including a dataset of natural images (presumably whose power spectrum is vastly different than that of MNIST) is important (especially given that as Stringer et al showed the dimensionality of the data governs the power law). At the very least, show the power spectrum of the raw MNIST data in the appendix. Also, show the spectra! I found myself continually having to refer to the appendix to look at the actual spectra of the different networks. Given that they are of central importance to the paper, I think it would be better if as many of these actual spectra could be moved to the main text.

Correctness: The range of beta values chosen do not seem to strongly affect the spectrum (Fig A6). Consider using a wider range?

Clarity: Yes

Relation to Prior Work: Yes

Reproducibility: Yes

Additional Feedback: Line 233: s/affect/effect Typo in Fig 1 caption: "and _is_ also robust"


Review 4

Summary and Contributions: This paper involved the theory of 1/n power-law in neural representation to improve the robustness of artificial neural networks to adversarial attacks. The authors have explored whether “1/n neural code” can make neural networks more robust, and how the intermediate representation of networks change with different layer configuration. Experimental results showed the spectral regularizer proposed based on the “1/N neural code” can play a role in MLP and deep neural networks under the threat of FGSM and PGD.

Strengths: - The authors proposed a novel spectral regularizer to reveal the benefits of the 1/n power-law found in neural representations. - Compared with vanilla networks, the spectral regularizer could help improve the robustness of MLP and DNNs in some cases.

Weaknesses: - The spectral regularizer lags far behind the Jacobian regularizer in performance, and it also fails to improve the robustness of models in some cases. For example, as shown in Figure 5, the Jacobian regularizer could help to increase the robustness of MLPs and CNNs with the strength of the adversary became larger, while the spectral regularization and vanilla networks will become failed. - It seems that the second question asked in the introduction didn’t be answered in the following of the paper.

Correctness: Correct. The authors based on the 1/n neural representation found by Stringer et al in mouse visual cortex, proposed a regularizer to enforce the power-law decay on the hidden layers.

Clarity: There are also a lot of grammatical errors.

Relation to Prior Work: Yes. This paper wants to introduce the representation phenomenon into the artificial neural network, and analyze the generalization of artificial neural networks from the perspective of kernel spectral law. However, as the effectiveness of spectral regularizer was not better than the prior regularizer, i.e. the Jacobian regularizer, the authors may need to explain other goodness of the novel regularizer compared with the previous ones.

Reproducibility: Yes

Additional Feedback:

[Author Response · NeurIPS 2020]

We thank the reviewers for their insightful comments and suggestions. We first reiterate the main goal and contributions. We asked (a) if having a $1/n$ neural code make neural networks more robust, and (b) how does the neural code employed by the intermediate layers affect the robustness. To answer these neuroscience-inspired questions on neural representation, we developed a pedagogical spectral regularizer that encourages an $1/n$ eigenspectrum on artificial networks. We demonstrated that networks with a $1/n$ eigenspectrum were more robust (sec 4.1). We provided empirical evidence that the neural representation employed by intermediate layers have a drastic affect on the robustness of the network regardless of the eigenspectrum of the last layer (sec 4.2, 4.3). We emphasize that the goal of this work is not to get SOTA performance on adversarial robustness or on other computer vision tasks, nor to design a practical training scheme. Rather, **our analyses elucidate the role of $1/n$ eigenspectrum observed in biological neural networks and also serve as inspiration for the design of future deep learning architectures; this is an exciting avenue of research that we leave for future work**.

As pointed out by R1,R2 & R3, our experiments were only run on MNIST. This was a deliberate choice as using MNIST has many advantages: 1) its simplicity makes it easy to design and train highly-expressive DNNs without relying on techniques like dropout or batch-norm, and 2) the models were able to be trained using a small learning rate, ensuring the efficacy of the training procedure detailed in section 3.2. This allowed us to isolate the effects of a $1/n$ neural representation, which may not have been possible if we use a dataset like CIFAR-10. We agree that the results would be bolstered if it were run on a natural image dataset like CIFAR-10 and leave that for future work but we note that the power-law behavior in rodents found by Stringer et al. was insensitive to the particular statistics of the input visual stimuli and was instead determined by the manifold dimension of the input. While the manifold dimension, $d$, of natural images is most likely larger than that of MNIST, both have manifold dimension much larger than 1; thus $\alpha = 1 + 2/d \approx 1$ for both MNIST and a dataset of natural images.

Per R1, we evaluated the robustness of networks on white-noise corrupted images where, in the interest of space, we only showcase the results of the networks from section 4.3 (Fig. 1). Having an $1/n$ eigenspectrum leads to an increase in robustness compared to their vanilla counterparts and for CNNs leads to **networks that are more robust than the Jacobian-regularized networks.** We would like to draw the attention of R5 to this particular case.

@R1: *"generic whitening is used for [. . . ] deep networks, while batch normalization (BN) is only used for the shallow network . . . "* We apologize for the confusion. To clarify, the whitening employed in section 4.2 is used to investigate the importance of intermediate representations. Whitening leads to a flat eigenspectrum which is the worst case scenario under the theory of Stringer et al., thus we whitened only the second hidden layer in the networks to see how this would affect the robustness of the network. Only the networks that end with a -Wh in figure 4 are the ones whose intermediate neural representation was whitened. BN was only used for the shallow neural networks in section 4.1 as we found that it helped the spectrally-regularized networks reach $1/n$ faster than the networks without. We will make sure to clear this up in the updated manuscript.

@R2: *"Evaluating the networks on other vision tasks."* We agree that evaluating the networks on other computer vision tasks is interesting but it is outside the scope of the paper and we leave it for future work. @R2: *". . . puzzled by the choice of regularization function . . . "* According to the theory developed by Stringer et al., having $\alpha < 1$ leads to undesirable properties. Thus, the goal of the regularizer is to get the eigenspectrum as close as possible to $1/n$ without going over, and the second term in the regularizer was added to heavily penalize eigenspectrum with an $\alpha < 1$.

Figure 1: $1/n$ neural representation leads to robustness against **white-noise corrupted images**. The SpecReg networks correspond to the networks shown in section 4.3 where all the hidden layers were regularized. SpecReg is more robust than the Jacobian regularizer for the CNN.

[Meta-Review · NeurIPS 2020]

The reviewers and I found this paper to be well-motivated and the question of how the spectrum of the covariance relates to robustness is one that the community would surely be interested in. There was some discussion regarding the methodology employed to investigate this question. First, there was general agreement that experiments on natural images would significantly improve the paper, especially because the spectrum of the input would be materially different. I do not believe that it is appropriate to relegate this experiment to future work, and I am not convinced by the rebuttal that including experiments with natural images would be computationally prohibitive. For example, downscaling or subsampling from CIFAR-10 would be a realistic option if the computations were too onerous on the full dataset. While I do not regard this analysis as mandatory for acceptance, I would strongly encourage the authors to include it in a revision. Second, there is some arbitrariness in the specifics of the regularization method and how it was implemented. More focus should be given to some of the details and how they do/do not affect the results. I am specifically concerned about the batch size and the effect of freezing the eigenvectors. Even some empirical justification for the chosen method/configuration would go a long way. Again, this analysis is not mandatory for acceptance, but including it would significantly enhance the takeaways that readers could draw from this paper. Overall, this is a borderline paper, but I think it just crosses the bar, and would be a strong paper if the authors included the above recommendations into the final version. So I recommend acceptance.